# Exercise-Associated Hyponatremia in Marathon Runners

**DOI:** 10.3390/jcm11226775

**Published:** 2022-11-16

**Authors:** Mark Klingert, Pantelis T. Nikolaidis, Katja Weiss, Mabliny Thuany, Daniela Chlíbková, Beat Knechtle

**Affiliations:** 1Department of Cardiovascular Surgery, University Hospital Bern, 3010 Bern, Switzerland; 2School of Health and Caring Sciences, University of West Attica, 12243 Athens, Greece; 3Institute of Primary Care, University Hospital Zurich, 8091 Zurich, Switzerland; 4Centre of Research, Education, Innovation and Intervention in Sport, Faculty of Sport, University of Porto, 4200-450 Porto, Portugal; 5Centre of Sports Activities, Brno University of Technology, 61669 Brno, Czech Republic; 6Medbase St. Gallen Am Vadianplatz, 9000 St. Gallen, Switzerland

**Keywords:** marathon, runners, hyponatremia, sports medicine, exercise, risk factors, epidemiology, review

## Abstract

Exercise-associated hyponatremia (EAH) was first described as water intoxication by Noakes et al. in 1985 and has become an important topic linked to several pathological conditions. However, despite progressive research, neurological disorders and even deaths due to hyponatremic encephalopathy continue to occur. Therefore, and due to the growing popularity of exercise-associated hyponatremia, this topic is of great importance for marathon runners and all professionals involved in runners’ training (e.g., coaches, medical staff, nutritionists, and trainers). The present narrative review sought to evaluate the prevalence of EAH among marathon runners and to identify associated etiological and risk factors. Furthermore, the aim was to derive preventive and therapeutic action plans for marathon runners based on current evidence. The search was conducted on PubMed, Scopus and Google Scholar using a predefined search algorithm by aggregating multiple terms (marathon run; exercise; sport; EAH; electrolyte disorder; fluid balance; dehydration; sodium concentration; hyponatremia). By this criterion, 135 articles were considered for the present study. Our results revealed that a complex interaction of different factors could cause EAH, which can be differentiated into event-related (high temperatures) and person-related (female sex) risk factors. There is variation in the reported prevalence of EAH, and two major studies indicated an incidence ranging from 7 to 15% for symptomatic and asymptomatic EAH. Athletes and coaches must be aware of EAH and its related problems and take appropriate measures for both training and competition. Coaches need to educate their athletes about the early symptoms of EAH to intervene at the earliest possible stage. In addition, individual hydration strategies need to be developed for the daily training routine, ideally in regard to sweat rate and salt losses via sweat. Future studies need to investigate the correlation between the risk factors of EAH and specific subgroups of marathon runners.

## 1. Introduction

Exercise-associated hyponatremia (EAH) was first described as water intoxication by Noakes et al. in 1985, who observed that athletes attending endurance events longer than seven hours developed hyponatremia. Since then, EAH has become an important topic in the field of endurance sports [1,2]. However, despite progressive research, neurological disorders and even death, hyponatremic encephalopathy continues to occur [3,4].

Hyponatremia occurs when the blood sodium concentration drops below 135 mmol/L (129–134.9 mmol/L) and a severe degree of EAH is typically < 125 mmol/L, which are both associated with signs and symptoms [5,6] Causes of EAH are usually individual sweat loss [7], excessive intake of (low-sodium or hypotonic) fluids [8], and possible hormonal imbalances [9,10], which occur more frequently at longer competitive distances [11]. However, cases of hyponatremia can also be found outside extreme sports, for example, in team sports and rowing, shorter races, and yoga [12]. Causes of EAH are usually individual sweat loss, excessive fluid intake, and possible hormonal imbalances, which occur more frequently over longer distances [13].

Among long-distance events, the marathon is one of the most symbolic races globally and has increased participation of all age groups and both sexes [14,15]. For instance, The TCS New York City Marathon, named after the major sponsor Tata Consultancy Services, is the premier event of New York Road Runners (NYRR) and the largest marathon in the world. More than 1.2 million people have finished the race since its first occurrence in 1970 [15]. Although runners competing in marathon events are also vulnerable to EAH, the aetiology of electrolyte imbalances in marathon runners is not well-researched [16]. Recently, this topic gained the attention of different research groups. McCubbin et al. concluded from their recent study about modeling sodium requirements that targeted sodium replacements may be unnecessary in elite marathoners [17]. Moreover, Fitzpatrick et al. discovered in a 9-year retrospective study an association between a collapse and serum creatinine and electrolyte concentrations in marathon runners [18]. Furthermore, evidence synthesis regarding the risk factors for EAH is warranted. The recommended sodium intake for young adults is 1.5 g per day to cover sodium sweat loss in non-acclimated individuals [19]. As described above, there is recent evidence that sodium replacement in marathon runners may not have any effect [17]. However, an intake of excess sodium can lead to methemoglobinemia and hypotension; thus, supplementation should be adjusted individually.

Besides limited evidence regarding EAH for endurance athletes that has been extensively reviewed by Rosner et al. [9], little information is available on marathoners. The present narrative review sought to evaluate the available evidence, and prevalence of EAH among marathon runners and to identify associated etiological and risk factors. Furthermore, based on current evidence, the aim was to derive preventive and therapeutic action plans for marathon runners. This topic is also important for marathon runners and stakeholders (athletes, coaches, medical staff, and event organizations) to better delineate training habits and fluid consumption during long-distance events.

## 2. Materials and Methods

The study was designed as a narrative review. The relevant literature was searched using a predefined search algorithm [20,21]. The selected articles were related to EAH in marathon runners and were published until June 2022. The search was conducted for sources of high-quality scientific information using three of the most widely used information databases in the area of health and sports sciences—PubMed, Scopus, and Google Scholar [22]. A combination of medical subject headings (MeSH) and free-text words was employed in the search [23]. The keywords used in the search were: (“marathon run” OR “exercise” OR “sport”) AND (“EAH” OR “electrolyte disorder” OR “fluid balance” OR “dehydration” OR “sodium concentration” OR “hyponatremia”). We found an overall number of 432 articles in the database of PubMed (Clinical Trials and Randomized Clinical Trials), 220 papers on SCOPUS (title search), and 1070 sources within Google Scholar (title search). In the end, a total of 135 articles were considered. Animal studies, in vitro studies, articles not published in German or English, and articles not related to the topic were excluded after the first screening of the search results (title and abstract screening).

## 3. Results

### 3.1. The Role of Sodium in Exercise-Associated Hyponatremia

In addition to making up the major cation of extracellular fluid, sodium also serves as an osmotic determinant in determining the extracellular fluid volume and plasma volume [24]. It is estimated that 95% of the total sodium in the human body is found in the extracellular fluid [25]. Additionally, sodium plays an essential role in cell membrane potential and the active transport of molecules across the cell membrane [26]. An active, energy-dependent process is required to maintain such a concentration gradient within the cell since sodium concentration inside the cell is typically less than 10% of that outside the cell membrane [27]. Clinically, dysnatremia is one of the most prevalent electrolyte disorders [28]. In the largest prospective study of dysnatremia conducted on athletes completing a marathon in Asia, no case of symptomatic dysnatremia was found. Runners with hypernatremia performed better, drank less water after the race and were better-prepared for the race [29]. In addition, Lüning et al. reported no case of hyponatremia, although the incidence of severe exercise-associated collapses was 1.53 per 1000 starting runners [30]. Dysnatremia can be divided into hypernatremia, meaning too much sodium in the extracellular space (>145 mmol/L), which results in water efflux from the cell and leads to cell shrinkage, and hyponatremia referring to water influx into the cell and results in cell swelling, which can become life-threatening. The term hyponatremia refers to a serum sodium level below 135 mmol/L [6,31].

In general, hyponatremia that occurs within 48 h is considered acute hyponatremia [32], a risk especially for patients in the postoperative period, patients with polydipsia and high physical activity [3,12]. The body has several regulatory mechanisms for pathological volume fluctuations or osmolarity changes [33]. Firstly, the volume regulation runs through the carotid sinus [34]. This is a vessel wall dilation in the internal carotid artery with baroreceptors, which, when the set point is exceeded, results in activation of the vagotonic via glossopharyngeal-induced stimulation of the cardioinhibitory neurons of the medulla oblongata, which initiates a reduction in blood pressure [34]. The physiological interplay of different mechanisms is critical to maintain homeostasis even under extreme conditions such as a marathon run.

Moreover, cardiac dilatation releases atrial natriuretic peptide (ANP), which reduces water retention in the body by balancing the osmotic gradient in the tubular system, thus lowering water balance [35]. The juxtaglomerular apparatus of the kidney regulates the release of renin [35]. When arterial blood volume decreases, renin is released, and the renin–angiotensin–aldosterone system (RAAS) is activated [36]. This causes the release of aldosterone via angiotensin I and II [36]. Aldosterone causes increased sodium reabsorption in the body and is associated with water retention, which causes the blood volume to increase again [36]. Conversely, osmoregulation runs through the hypothalamus, which releases the primary antidiuretic hormone vasopressin (AVP) in response to hypovolemia or increased serum osmolarity, which—in turn—inhibits water excretion in the kidney [37]. In the context of exercise, both the mechanisms of fluid intake and the hormonal regulation of vasopressin are responsible for maintaining serum sodium levels [38].

Hyponatremia can be categorized etiologically into four forms: pseudohyponatremia, euvolemic, hypovolemic, and hypervolemic hyponatremia (Table 1) [39]. In pseudohyponatremia, the plasma’s lipid, protein, and glucose content are increased, whereas the sodium content of plasma water is within the normal range (135–145 mmol/L) [39]. In euvolemic hyponatremia, total body volume increases, whereas total sodium is normal (135–145 mmol/L) [40]. This may be triggered by water-retentive drugs, hypercortisolism or the syndrome of inadequate AVP secretion (SIAVP) [40]. In hypovolemic hyponatremia, the volume deficiency is triggered either through the kidneys by diuretic drugs, an adrenocortical insufficiency, or extrarenal by vomiting, diarrhea or pancreatitis [32]. In athletes, including marathoners, hypovolemic EAH is triggered when exercising >20 h in a hot environment and/or with higher sweat sodium losses [41]. The crucial form in this work is hypervolemic hyponatremia, which can result from the overconsumption of fluids, likely in combination with non-osmotic stimulation of AVP secretion [42], and damage to the heart, liver, and kidney [43]. Due to the increased extracellular fluid with a decreased serum sodium concentration, macroscopic edema formation can occur [44,45].

EAH can develop symptomatically or asymptomatically, mostly during or within 24 h after exercise or competition [3,42]. EAH represents an acute form of hypervolemic or hypovolemic hyponatremia [3]. It has been confirmed that EAH is a serious condition associated with prolonged physical activity, and the complex pathophysiology of EAH is influenced by several factors [48,49]. When excessive water or hypertonic beverages are consumed, diluted hyponatremia occurs due to an overabundance of total body water compared to total exchangeable sodium in the body [47]. It has been proposed by Noakes et al. that EAH occurs as a result of excessive hydration during exercise, retention of excess fluid as the result of inadequate suppression of antidiuretic hormone production, and inactivation of osmotically circulating sodium or inability to mobilize osmotically inactive sodium from internal reserves [47]. EAH is also linked to rhabdomyolysis, platelet activation, cardiac problems, and renal failure [50,51,52,53,54,55].

The problem is identifying athletes with an inappropriate secretion of AVP or increased body water due to increased drinking [38]. It has been observed that some athletes achieved adequate diuresis to avoid gaining weight or, if they do gain weight, to maintain their serum sodium levels by mobilizing sodium from their internal stores as needed [38]. The diuresis that occurs in other athletes is inadequate, resulting in fluid retention, the release of ANP, and the expression of EAH [38,56]. Despite the fact that EAH may occur during dehydration [57], the majority of the cases of EAH that are symptomatic are those that are associated with overhydration when weight gain has occurred or inadequate weight loss has occurred during physical activity [9,47]. It should be highlighted that dehydration refers to the loss of body water from the interstitial and intracellular compartments associated with hypertonicity. Its clinical signs include altered skin turgor, dry oral mucous membranes, sunken eyes, inability to spit, and sensation of thirst [58].

EAH may also be caused by increased sympathetic nervous system activity, specific genotypes, and the RAAS system, as well as an increase in the splanchnic perfusion [7,59].

There is still some uncertainty regarding the role that sweating plays in the pathogenesis of EAH, as well as the possibility that urinary sodium loss may produce increased levels of atrial natriuretic peptide (ANP) and brain-type natriuretic peptide (BNP) in the physiology of EAH [42]. Nonspecific symptoms include confusion, headache, lethargy, seizures, anorexia, and even coma [12,47,60].

Marathon running is highly popular with an increase in the number of races and participants, especially the number of master marathoners and female runners [14]. Many marathon runners suffer from gastrointestinal disorders due to dehydration and electrolyte imbalance [61,62,63,64,65]. The reduced blood volume during exercise may reduce blood flow to the intestinal region [22]. There is a possibility that this may interfere with the digestion system’s normal ability to secrete/absorb nutrients [63,66]. A runner may also experience gastrointestinal symptoms when consuming non-steroidal anti-inflammatory drugs (NSAIDs) [67]. EAH in marathon runners and its associated risk factors will be more deeply explored in the next section. As confirmed by Rüst et al., EAH tends to be a complication that occurs during longer distances [68]. This is also indicated in a study by Hiller, who examined the frequencies of EAH of a short race and that of an Ironman and showed that EAH only occurred in the Ironman [13,68]. It is difficult to make an accurate statement regarding the frequency because different studies report different prevalence regarding the occurrence of EAH. According to the literature, the prevalence of EAH in marathon runners ranges from <1 to 22%, translating to a clinically significant number of marathon runners affected by EAH (Table 2). The mean and standard deviation of prevalence based on these data was 8.2 ± 7.9%. This high prevalence and burden in marathon sports justify an in-depth analysis of risk factors and strategic treatment algorithms for this entity to lower the prevalence in the future.

### 3.2. Aetiology and Evidence of Exercise-Induced Hyponatremia in Marathon Runners

The often-cited comparative study by Noakes et al. collected data from competitions between 1997 and 2004, including the 2002 Christchurch New Zealand marathon, 2002 Boston marathon, and 2003 Houston marathon, and created a representative sample size of 2135 runners [47]. Their main objective was to relate serum sodium concentration changes during a race to increased fluid intake measured by body-weight change. For this purpose, blood values and weight changes were retrospectively analysed. The data were divided into three categories: overhydration, euhydration, and dehydration. The serum sodium concentration was divided into hypernatremia, normonatremia, asymptomatic hyponatremia, and clinically significant hyponatremic states.

Among the athletes who were overhydrated, 44 suffered from biochemical hyponatremia (serum sodium 129–134.9 mmol/L) and 25 from clinically noticeable hyponatremia (serum sodium < 128.9 mmol/L). From the euhydration group, 41 subjects developed asymptomatic hyponatremia, and 6 developed symptomatic hyponatremia. No dehydrated individuals developed symptomatic hyponatremia, whereas 38 dehydrated runners experienced asymptomatic hyponatremia. Percentagewise, six percent of the athletes experienced asymptomatic hyponatremia without symptoms, and one percent experienced a symptomatic form.

Either euhydrated (39%) or with weight loss (50%), 89% of the athletes completed these events with normal (80%) (135–145 mmol/L) or elevated (13%) (>145 mmol/L) serum sodium. The serum sodium levels of 231 athletes with weight gain were normal or above the normal range at 70%, 129 to 135 mmol/L for 19%, and below 129 mmol/L for 11%. Of the 11% who gained weight during the race, 30% experienced hyponatremia (19% biochemical (129–135 mmol/L) and 11% clinically significant (<129 mmol/L) EAH). The relationship between post-race serum sodium levels and weight change during the race could be represented by a linear function with a negative slope. A significant difference was observed between the overhydrated and the euhydrated or dehydrated groups concerning serum sodium concentration. Serum sodium levels were also significantly different between the euhydrated and dehydrated groups. The risk of developing EAH first existed at a body-weight loss of −6 to −2% and increased to 10% at a body-weight change of −2 to 0%. Thereafter, the risk increased gradually so that those with a 4% weight gain had an 85% chance of developing EAH [47].

Similarly, Glace et al. reported an inverse correlation between changes in body weight and serum sodium concentration in a prospective observational study involving 26 carefully selected trained ultra-marathon runners [75]. Only one case of a temporary drop in sodium level below 135 mmol/L was documented. Urinary sodium excretion decreased during the race, while plasma volume increased by 12% toward the end [75]. The characteristics of the studies from Noakes et al. [47] and Glace et al. [75] are shown in Table 3.

The observational study by Reid et al. examined haematological and biochemical parameters during a marathon, incidence rates, and the change in parameters with the use of analgesics in 134 athletes [76]. They found an average weight loss of 1.6 kg in men compared with 0.9 kg in women. The serum sodium concentration after the race was significantly higher in men than women. Percentage weight loss, accounting for the lower starting weights of women, was also significantly different between the two groups. It has been found that serum sodium concentration and body-weight change are significantly correlated. Further, sodium concentration decreased by 1.16 mmol/L per hour of finishing time, with a statistically significant inverse correlation between serum sodium concentration and finishing time. However, none of the athletes developed hyponatremia [76].

Hew et al. showed that 0.31% of the entrants of the Houston Marathon presented to the medical area with hyponatremia [69]. Specifically, excessive fluid consumption and prolonged finishing times were the primary risk factors, and vomiting was the only clinical sign distinguishing hyponatremia from other conditions that cause exercise-associated collapse. In contrast, another study did not find evidence for clinical signs identifying runners at the end of a marathon [58].

On the other hand, among healthy athletes who ran a marathon in cold temperatures, a significant proportion (12.5%) developed asymptomatic hyponatremia [77]. It has been found that runners who were not hyponatremic consumed more fluids and gained more body weight due to increased fluid consumption. According to Hsieh et al., clinically relevant hypervolemic hyponatremia was reported in 5.6% of the marathon runners who required medical treatment [72]. Cheuvront et al. demonstrated that when ad libitum fluid intakes replace ~60–70% of sweat losses, runner thermoregulatory responses are maintained [78]. Hypohydration was observed to have a paradoxical positive relationship with sweat rate in a 42 km marathon race, while liquid intake had a negative relationship with sweat rate [79]. This may be explained by decreased heat conductance, which, in turn, raised the core temperature, causing sweating to increase [79]. According to Roberts et al., an urban 42 km marathon revealed that 59% of runners presented with exercise-associated collapse during the race in a study where, of 81,277 runners, 112 received intravenous fluids and 30 were transported to a hospital for treatment [80]. It has also been shown that non-cardiogenic pulmonary edema can lead to hyponatremic encephalopathy in healthy marathon runners [60].

Mettler et al. assessed data from 167 athletes to understand the mechanism of development of EAH during a marathon [74]. The hypothesis was that the prevalence of hyponatremia would be lower than in longer distance studies because the time available for over drinking would be shorter. They determined the body weight, serum sodium levels, and osmolality before and after the race. Five runners developed asymptomatic hyponatremia, including four women and only one man. Female runners had less prior competitive experience, lower body mass index (BMI), and slower running time than their male counterparts. Weight changes correlated with post-race serum sodium levels and relative sodium concentration changes during the marathon. Fluid intake correlated significantly with sodium change during exercise [74]. A study on 21 marathon runners that developed EAH found that pre- to post-race weight loss negatively correlated with fluid consumption and finish time [70]. The same study advised a weight loss >0.75 kg of body weight during the marathon to effectively reduce the risk of EAH occurrence [70].

Using different variables, such as body weight, body composition, running speed, weather conditions, and sweat sodium concentration, Montain et al. developed a mathematical model to predict the effects of drinking on hydration status and plasma sodium concentration [81]. Based on the predictions, it is anticipated that participants will be adequately hydrated over a 42 km distance if the fluid intake recommendations of the International Marathon Medical Directors Association (IMMDA) are followed. The mathematical simulations showed the complexity of defining fluid and electrolyte consumption rates during athletic competition.

In 2011, a clinical study on ultra-distance marathon runners by Knechtle et al. considered anthropometric data, training history, and race experience in addition to the determination of body weight, blood values, and urine values [82]. All parameters collected were compared among target runners. Only 7% of participants developed EAH. Hyponatremic runners were not slower than non-hyponatremic athletes. There was no difference between athletes regarding training and prior race experience. This study also showed a correlation between body-weight change with post-competition sodium concentration and relative sodium changes during competition. Ultra-marathon runners with EAH did not drink more compared with athletes without hyponatremia. Fluid intake was not associated with end-of-race sodium concentration and sodium changes [82].

In the observational study by Stümpfle et al., eight runners were included in a race in cool outdoor temperatures [83]. Weighing and measuring serum sodium levels were conducted prior to and following the race. Dietary habits during the race were analysed to determine fluid and sodium consumption. The athletes who developed hyponatremia weighed less than the normonatremic group before and after the race. Significant weight reduction was noted only in the athletes without hyponatremia. The hyponatremic group consumed more fluid (0.1 litres per hour more) and 60 mg less sodium per hour than the normonatremic group on a nutritional level [83].

In a prospective observational study, Almond et al. analysed data on demographics, training history, fluid intake, blood counts, and urine output in 488 marathon runners. Asymptomatic EAH was found in 13% of the athletes and severe hyponatremia in 0.6%. Using univariate analysis, the development of EAH was associated with weight gain, fluid intake greater than three litres, running longer than four hours, and low BMI. Multivariate analyses revealed a relationship between hyponatremia, weight gain, running time exceeding four hours, and high BMI. During the race, 35% of the runners gained weight. There was an association between fluid intake of three or more litres, fluid intake per mile, longer running time, female gender, and a BMI under 20 kg/m^2^ [71]. Hoffman et al. conducted an observational cohort study in which 47 athletes participated in a marathon event. Of 104 study participants, 47 were able to complete the race. During this race, 30% of the 47 participants developed EAH. Post-race sodium concentration and percent change in body weight were directly related. However, EAH showed no association with athlete age, sex, achieved running time or NSAIDs use. This observation is highly relevant as intake of NSAIDs has been indicated as a risk factor in the development of exercise-associated hyponatremia [84,85]. Athletes with EAH had less running experience than athletes with sodium levels in the normal range. In addition, 7 of the 14 hyponatremic athletes were found to have lost 3–6% of their body mass [86]. Reid et al. found no cases of hyponatremia in runners completing a marathon. Notably, serum potassium concentrations were significantly higher in runners who had taken NSAIDs [76].

The prospective observational cohort study by Scotney and Reid examined serum sodium levels, weight changes, and hydration schedules of 41 ultra-distance mountain runners [87]. The subjects had completed an average of four endurance events over the past five years. A total of 39 percent of the participants drank a set volume per unit of time, 27% drank according to thirst sensation, 12% drank at every hydration station, 2% drank at some hydration stations, and 20% reported other drinking schedules. Among the athletes, only one developed asymptomatic EAH. This individual drank according to thirst sensation and consumed 5.3 litres of water and electrolyte solution in a 9:1 ratio. Body weight was identical before and after the competition. In addition, it shows that ~150 mg of diclofenac was consumed 24 h before the end of the run. There were no statistically significant differences in age, biochemistry, running time, and percent weight loss for pain medication (NSAIDs) users compared to non-users. It was found that changes in serum sodium concentration were significantly correlated with body-weight changes. There was an inverse relationship between hydration and serum sodium concentration. Runners with hydration protocols had lower blood sodium concentrations than athletes who drank according to thirst perception and, on average, managed with less fluid intake [87]. It is possible that maintaining one’s body weight by drinking may negatively affect exercise performance, especially for those who run slowly. Drinking may also increase the likelihood of hyponatremia associated with exercise, especially for those who run slowly [88].

Lebus et al. investigated the association of decreasing serum sodium levels with weight changes, total body water (TBW) changes, and extracellular volume (ECW) changes in 45 ultrarunners during a race [89]. Body weight and serum sodium concentrations decreased significantly by 2–3% from start to finish. In contrast, BFW and ECW were unchanged. In addition, a significant relationship was observed between finishing time and percent change in body mass, BFW, and ECW. Neither body-weight change nor finishing time was associated with post-race serum sodium concentration. Another study observation was that nine runners with serum sodium levels below 135 mmol/L lost ~3.58% of their body weight. EAH had occurred in 51.2% of the runners [89].

As part of their investigation of risk factors for EAH among marathon runners, Chorley et al. conducted a prospective observational study in which serum sodium levels, body weight, and fluid intake were measured at different time points in 96 marathon runners [74]. Taken together, male participants completed four times as many marathons beforehand as the female participants. Serum sodium levels decreased in 87% of all participants. During the race, there was a significant drop in serum sodium concentration, resulting from a drop in serum sodium concentration prior to the marathon, weight loss during the race, and excessive fluid intake. High fluid consumption correlated with longer running time, male gender, and warm ambient temperatures. Lower body-weight loss correlated with lower body weight at the beginning of the race, higher fluid consumption, and longer finishing time. Losing less than 0.75 kg of body weight increased the risk of developing hyponatremia 7-fold. Women consumed significantly less fluid than men and had an even lower sweating rate, resulting in an overall positive fluid balance. Although these differences between men and women were significant using the simple *t*-test, using general linear models for several variables, sex did not affect post-race sodium concentration, sodium concentration change or weight change [70].

Since the studies cited so far did not always clearly show specific sex differences, the following four studies explicitly address the question in more detail. In the clinical study by Wagner et al., 25 male and 11 female athletes were examined for prevalence cases, including weather conditions and water temperature [90]. They determined anthropometric data and competition experience in addition to training data and a blood and urine sample before and after the marathon. The analysis of anthropometric data and training performance showed that males were taller and heavier and had less fat mass and more skeletal muscle mass than females. Analysing pre-race experience, training, and energy, fluid, and electrolyte intake during the marathon, there was no sex difference between the sexes in energy, fluid, and electrolyte consumption. There was barely a difference in speed between men and women. There was a correlation between changes in body weight and changes in plasma sodium concentration following a race. Plasma sodium concentration post-race or changes in plasma sodium concentration were not related to finishing time. Post-race plasma sodium showed no difference between the sexes. Two of the 25 men and 4 of the 11 women developed EAH. Salt or fluid intake was not associated with plasma sodium changes in either sex [90].

In the observational study by Eijsvogels et al., fluid balance in endurance athletes was examined concerning sex differences [91]. Using 98 athletes, including 56 men and 42 women, body-weight changes, heart rate, body temperature, and haematological parameters were collected. Subjects were asked to run at their self-selected speed and at 69 ± 11% of their maximum heart rate. Running time and intensity was comparable between the genders. While baseline and maximum core body temperature were significantly higher in women, the increase in temperature was comparable between the two sexes. Males were found to have higher weight fluctuations than females and a greater risk of dehydration. As stated in a previous study analysing data from the Houston marathon, women were found to hydrate more than men during a marathon run [92]. A backward logistic regression analysis revealed that only gender and fluid intake significantly contributed to the occurrence of 2% body mass loss. Twenty-seven percent of men developed hypernatremia, whereas women did not. Men also consumed less fluid during the race than women. Exercise-induced changes in plasma sodium levels did not differ between premenopausal and postmenopausal women [91]. It is noteworthy that marathon runners frequently and consistently reported high rectal temperatures following their race [93]. However, Myhre et al. reported that middle-aged runners who successfully completed the marathon in a warm environment maintained a steady-state heart rate and rectal temperature [94].

As an indicator in marathon runners, the post-race rectal temperature is predicted by the metabolic rate rather than dehydration [95]. A large portion of the energy produced during endurance exercise is in the form of heat, which cannot be stored. Consequently, marathon races are generally scheduled at cooler times of the year or day in order to ensure adequate heat loss to the environment [96]. A recent study suggested that hyperthermia may directly affect the brain, which could contribute to fatigue during prolonged exercise in a warm environment [97]. The comparative study by Lara et al. determined the individual differences in electrolyte concentrations in sweat in marathon runners [98]. Here, sweat electrolyte concentration was determined in 157 athletes using sweat patches. In addition, body-weight changes were measured before and after the marathon. Sweat from women contained lower concentrations of sodium and chloride than sweat from men. In addition, sweat production and running speed were also lower. The study participants could be classified according to the salt concentration in sweat. A total of 26.8% of the study population was classified as “low salt sweat producers”, 53.5% of the sample was classified as “typical sweat producers”, and 19.7% of the subjects were classified as “salty sweat producers”. The group of salty sweat producers consisted exclusively of male marathon runners. A significant correlation between sodium concentration and running speed was demonstrated, but the association between sweat sodium concentration and running speed was weak and negatively correlated. There was no correlation between sweat electrolyte concentration, athlete age, body characteristics, or prior training or competition experience [98].

The prospective observational study by Pahnke et al. analysed the influence of fluid intake and salt losses through sweating on serum sodium concentration in 46 runners [99]. Endurance-trained athletes were studied during a stationary cycling interval of 30 min at 70–75% of maximum heart rate in a warm outdoor laboratory at a wet bulb globe temperature of ±28.3 °C, 3–7 days prior to race day. Sweat rate was determined using body-weight changes and salt concentration in sweat. A series of measurements of body weight, sodium concentration, and nutrient intake was conducted prior to and after the race. In men, serum sodium concentration during the training was negatively correlated with sweating rate, sweat sodium loss, and change in body mass [99]. In women, however, serum sodium concentration correlated solely with body-mass change, and the rate of sweat sodium loss correlated with sodium intake in women only [99]. These findings indicate differences in fluid and sodium balance and interactions between sexes, but causal evidence is lacking [99].

The randomized controlled trial by Koenders et al. focused on dietary sodium restriction [100]. During exercise in the heat, water intake was equivalent to body-weight loss during exercise, and the objective was to determine whether dietary sodium influenced plasma sodium concentrations. Each of the subjects completed nine days on a low-sodium or high-sodium diet. All participants received the same breakfast before the start of the study. Furthermore, body weight was recorded, and urine was collected. All subjects were asked to perform an endurance exercise for three hours at 55% of their maximum oxygen uptake in the heat and constant convection at 34 °C. Meanwhile, distilled water adjusted for body-mass loss was offered every 15 min and a blood sample was taken every half-hour. Plasma sodium was decreased at low sodium intake before and throughout the exercise. Both diets resulted in a similar reduction in plasma volume during exercise. Sweat sodium concentration and sweat rate remained constant regardless of diet. Urinary sodium during and after exercise was five times greater with high sodium intake than with low intake. Five participants from the high-sodium group and only three from the low-sodium group were able to exceed the three-hour mark [100]. This study indicated that sodium intake may also be a performance-limiting factor and is not solely determined by sodium intake during exercise.

Hoffman and Myers discussed whether sodium intake during extreme endurance sports could be used as a preventative measure against EAH [101]. Two hundred athletes were measured before, during, and after the race to determine their serum sodium levels. Athletes needed to be weighed before, during, and after a race, along with reporting how much fluid and sodium they had consumed. Sodium intake was perceived by 93.9% of the athletes, and no difference in sodium intake was found between hyponatremic and normonatremic athletes. No significant differences in athlete characteristics were found between the two groups. There was a direct relationship between post-race sodium concentration and the frequency of sodium consumption from supplements, regardless of the timing of sodium intake.

The randomized controlled trial by Hew-Butler et al. aimed to determine in 413 runners whether sodium supplementation during a 226 km Ironman triathlon would provide consistent serum sodium levels and result in better performance [102]. Participants in the control group received 40 placebo tablets filled with 596 mg of starch. Subjects in the sodium supplementation group consumed 3.6 g of sodium in 40 identical-looking tablets during the race. It should be noted that food and liquid intake was allowed ad libitum and was not considered in the analysis. Regarding the haematological parameters or baseline characteristics of the athletes, there was no significant difference between the sodium and placebo groups. The researchers found that triathletes who consumed placebos or salt tablets ad libitum during exercise maintained their serum sodium levels between normal and low levels for an average of 12.5 h.

The randomized controlled trial led by Del Coso et al. used 26 runners to test the effectiveness of sodium supplements in endurance sports [103]. The experimental group received electrolyte supplementation with 2.6 g sodium distributed over 12 capsules, while the control group received cellulose. Prior to and following the race, measurements of body mass, maximum strength, and maximum jump height were taken, as well as blood samples. Compared with the control group, the intervention group significantly shortened its finishing time. Sweat loss and sweat sodium concentration were comparable in both groups. The intervention group with sodium intake experienced less weight loss (−2.8 to −3.4%) and higher sodium serum levels after the triathlon. In the control group, although sodium concentration increased from 141.4 mmol/L to 143.4 mmol/L, the increase was higher in the intervention group, from 141.8 mmol/L to 144.9 mmol/L at the end of the race. Thus, the study showed that supplemental sodium administration can affect blood sodium levels and body-weight loss.

The randomized trial by Anastasiou et al. was about the effectiveness of sodium-containing drinks in endurance races at temperatures of 30 °C [104]. Four groups were formed based on the level of training of the athletes. A carbohydrate-electrolyte drink was provided to Group 1 containing 36.2 mmol/L sodium, a carbohydrate-electrolyte drink was provided to Group 2, a carbohydrate-electrolyte drink was provided to Group 3, mineral water was provided to Group 4, and distilled water was provided to Group 4. Tests were conducted to measure the level of serum sodium, the level of plasma osmolality, the volume changes in the plasma, and the frequency of cramping in the muscles. There was no difference in plasma volume between the low-sodium and high-sodium groups after sodium supplementation. There was no difference in serum sodium between Group 1 and Group 2. In contrast to Groups 1 and 2, there was a large decrease in serum sodium in the groups without sodium addition [104].

In the prospective observational study by Twerenbold et al., the effects of sodium supplementation with different doses were examined in 13 women [105]. Body weight was determined, and a blood sample was taken before and after a 4 h run. Participants were instructed to document food and beverage intake during the 24 h prior to the study. In a four-hour running experiment, subjects were asked to run as many miles as possible. Three fluids with different sodium concentrations (low sodium, 0.68 g and 0.41 g sodium) were provided. Subjects were randomly divided into three groups in which the fluids were consumed in a different order. In the high-sodium supplementation group, plasma sodium levels decreased in 77% of the subjects. In the low-sodium diet, 92% of the women had lower levels after the race than before. In the mineral-water-only group, sodium levels decreased in all participants. There was a significant difference between the high sodium and water intake groups regarding the change in plasma sodium concentration over the whole trial. Specifically, the mean decrease in plasma sodium concentration was significantly lower in the high-sodium group compared to the water-intake group [105]. The authors recommend sodium replacement at 680 mg/h for women in case of fluid overload induced by fluid consumption during a longer run. This will help to reduce the risk of EAH development [105].

Barr et al. evaluated the need for sodium supplementation during endurance exercise for up to 6 h [106]. They studied three different intervention groups: water supplementation (W), sodium supplementation (S), and without-fluid supplementation (NF). Blood samples were collected at different time points during the run. Body weight and sweat analyses were obtained at the end of endurance training. Plasma sodium decreased in the W group and the S group, whereas plasma sodium increased steadily and was significantly higher in the NF group. Salt intake did not show significantly higher plasma sodium compared to water intake alone [106]. One study measured body weight, serum sodium levels, and plasma volume as a function of sodium supplementation in 38 athletes. Some athletes received salt tablets (0.7 g per tablet) to be taken during the race [107]. A second group, without salt tablets, served as a control group. Before and after the race, weights and blood samples were taken. The intervention group showed a greater increase in sodium concentration during the race, but this was not statistically significant (1.52 mmol/L versus 0.84 mmol/L) [108]. In addition, none of the athletes finished with a sodium concentration < 135 mmol/L. Overall, it can be concluded from the literature that hyponatremia is a serious and not infrequently occurring complication in endurance sports such as marathon running and that improved education of the athletes in this regard is needed.

Related to therapeutic measures in pre-existing EAH, two clinical studies were found that addressed EAH treatment—the study by Owen et al. [109], a randomized controlled trial with 26 affected athletes, and the study by Rogers et al. [110], which included 8 volunteer athletes. A 3% saline solution was administered orally or intravenously in both studies to examine the effects of oral versus intravenous administration on persistent EAH. Two measurement time points were collected—before the intervention and 60 min after saline administration. The serum sodium concentration before and after the intervention did not differ significantly between the intervention and control groups. Blood sodium increased over time in both intervention groups. Patients receiving the intravenous bolus had a greater mean plasma volume increase. In the prospective study by Rogers et al., a significant sodium concentration change was recorded in the intravenously treated group [110]. Within 60 min of intravenous administration, serum levels were increased by 4 mmol/L.

## 4. Discussion

### 4.1. Risk Factors

The first aim of the present work was to give an overview of evidence-based risk factors of EAH and to derive possible risk groups. The risk factors can be divided into event-related and person-related (Figure 1). Event-related risk factors include more than four hours of endurance efforts, frequent fluid availability, and competition conditions with high ambient temperatures. It is important to note that risk factors that are athlete-dependent include overdrinking, body-weight gain, low BMI, less prior endurance experience, and the use of NSAIDs, as well as female sex that is suggested to play a role, but causal evidence is still lacking [87].

An important risk factor for developing EAH is sustained excess fluid consumption that exceeds water loss through sweating, respiratory movements, and renal excretion, resulting in a positive fluid balance over time [47]. This positive fluid balance is thus reflected in body-weight gain or reduction in body-weight loss. In almost all cases of symptomatic EAH, body-weight gain or nearly unchanged body weight is seen during exercise [42]. The majority of the studies reviewed came to a consensus that body-weight changes were statistically significantly inversely correlated with serum sodium concentrations [47,74,75,76,82,87]. A correlation was also suggested by Almond et al. [71] and Chorley et al. [70], but neither study reached statistical significance.

Regarding the relationship between marathon race-induced changes in serum sodium and body weight, in a study conducted by Chorley et al. [70], lower post-race serum sodium concentrations were related to less body-weight loss in athletes. The results of Stümpfle et al. [83], Wagner et al. [90], Hoffman et al. [86], and Lebus et al. [89] contradicted the other studies. Wagner et al. [90] and Hoffman et al. [86] showed a correlation between body-weight loss and sodium concentration after a competition. A small sample size and a significantly longer exercise duration were two characteristics that distinguished the four studies with the unusual body-weight loss phenomenon. The body-weight loss in extreme endurance athletes presumably relied on high glycogen loading prior to competition. Since ~3 g of water are bound to each g of glycogen, glycogen depletion can cause body-weight loss while maintaining hydration [89]. It is also not uncommon for athletes in long races to arrive at the finish line in a dehydrated state [111].

Dehydration may also be responsible for body-weight loss [89]. At the end of a race, increased fluid consumption was significantly correlated with low serum sodium levels [47,74,84,87]. Further, the study results of Chorley et al. [70], Stümpfle et al. [83], and Almond et al. [71] supported this hypothesis [70,71,83]. On the contrary, fluid intake and serum sodium change were not associated in a study with 145 subjects [82]. According to Traiperm et al., well-trained and educated adolescent marathon runners may not experience clinically significant electrolyte and haematological changes [112].

Some authors pointed out that athletes should ignore a set hourly drinking rate and drink only when they feel thirsty [47,87]. However, in general, this important fact has still not been sufficiently integrated into the behaviour of most athletes. According to Noakes et al., the sports drink industry has taken the opposite interest and knowingly ignored knowledge about excessive fluid intake as a major risk factor for EAH since as early as 1985 [47].

A significant correlation between the female sex and post-competition serum sodium levels below 135 mmol/L has not been found [113]. However, some studies have shown that women consume significantly more fluids and have less body-weight loss compared to men [76,99,113]. At the same time, less sodium chloride in sweat and significantly lower sweat production overall have been demonstrated compared to their male counterparts [98,99]. Numerous studies have shown lower average body weight, less competitive experience, and lower running speeds for women [74,98,113]. Wagner et al. attempted to explain the higher incidences of EAH in women as the syndrome of inadequate AVP secretion occurring more frequently in females [90]. Due to their smaller body size and lower metabolic rate, women tended to sweat less than men [98]. This factor and a permanent fluid intake during exercise could lead to a very positive water balance. This effect is influenced by the release of the antidiuretic hormone [114]. During a marathon run, the hormone response may have stemmed from the effects of the exercise itself rather than dehydration [114].

A woman’s diuretic response to water stress may be compromised by estrogen and progesterone. These hormones interfere with the renal actions of antidiuretic hormone, leading to higher water turnover, which is determined by her menstrual cycle phase [90]. Inadequate prior race experience or training is among the risk factors for EAH [70,74]. In two studies, prior running experience was narrowed to two marathons in females, while the male sex recorded six to eight races [70,74]. Since in both cases the female sex showed an increased incidence of EAH, the lower race experience could offer an explanation [70,74]. It is, however, important to mention that there are other studies confirming no effect of sex on EAH development [115], indicating the need for further research to clarify potential sex-specific effects.

In addition, in the study by Hoffman et al., the hyponatremic runners had completed only one marathon in the past, while the normonatremic competitors had three to show for it [86]. At the end of a race, running times over four hours were significantly correlated with lower serum sodium levels, and analyses suggested that these two factors were directly connected [71,76]. According to one study, pain medication had no statistically significant impact on biochemical systems in the body, regardless of whether it was consumed by users or non-users [87].

Data on NSAIDs are conflicting, and further studies are necessary to determine if they are risk factors for the development of EAH, both in terms of classification and dosage [42]. However, it has been demonstrated that NSAIDs may directly or indirectly affect renal function [85,116,117,118]. In addition to dehydration during running, this may result in impaired renal function and kidney-stone development over time [119]. The extent to which outdoor temperatures directly influence the development of EAH could not be clarified within the scope of this work, as there was too little information on this and too much variation in the individual studies. For this reason, further controlled studies are required.

A not-insignificant risk factor investigated in the two studies conducted by Lara et al. is the sweat rate and the sweat electrolyte concentration in athletes [59,98]. They showed that ~20% of marathon runners had a high sweat sodium concentration of more than 60 mmol/L in their sweat. However, there was no direct correlation between sweat electrolyte concentration, sweat rate, physical characteristics, and competition experience explaining these interindividual variations. Only sex played an influence on the sweat production rate and composition. Females had less sodium in their sweat and a lower sweat rate [98]. Regarding sex differences, it can be summarized that the combination of “female gender” with low BMI and low, competitive experience increases the likelihood of developing EAH. Other findings from this narrative review are that there are no clear studies regarding risk factors. The most valid risk factor seems to be weight gain during endurance sports, most of which was associated with increased fluid consumption. It is difficult to establish clear risk factors because there are many components to consider [120].

### 4.2. Prevention and Clinical Implications

Prolonged continuous stress is associated with a considerable loss of fluids and NaCl [121]. In endurance sports, it is important to maintain physical performance through an adequate compensatory supply of electrolyte-containing fluids [122]. This review was conducted in order to highlight evidence-based prevention strategies. Many studies have examined sodium intake in endurance sports and, in most cases, concluded that sodium intake during endurance performance is associated with higher blood sodium concentrations [123]. As part of a study by Koenders et al., sodium levels were measured after the competition for athletes following high and low sodium diets, while the fluid intake was adjusted to allow for weight loss [100]. According to this study, a high sodium intake reduced cardiovascular and thermal stress and associated perceptions, suggesting that reducing sodium in the diet of endurance athletes should not be recommended.

As a result of the combined muscular and thermoregulatory blood circulation demands of running in a hot environment, runners experience high levels of hyperthermia, likely due to a cardiovascular overload due to the combined results of progressive dehydration [124,125,126]. A randomized controlled trial by Del Coso et al. comparing athletes with 2.6 g sodium intake to placebo supplementation concluded that ingestion of ~470 mg sodium per hour was associated with a run-time reduction and improved water and sodium balance [103].

The four-arm study by Anastasiou et al. examined the effectiveness of sodium-containing beverages relative to serum sodium concentrations [104]. It was found that even a moderate amount of sodium was sufficient to maintain plasma volume and attenuate the decrease in plasma sodium concentration following exercise. Through the results of the study by Twerenbold et al., a recommendation of 0.68 g/h sodium intake can be made for women [105]. In the study by Schrader et al., a carbohydrate-based fluid intake was compared to a tap-water-based fluid intake [127]. Sodium concentration decreased with water-only intake but was again stabilized with water-plus-maltodextrin intake [127].

According to Speedy et al., although sodium concentration increased significantly in the intervention group after supplementation with 0.7 g of sodium per hour, this increase was not statistically significant compared to the control group [108]. There was no evidence that sodium supplementation resulted in a significant change in serum sodium levels. The necessity of supplementation, however, has been questioned by some authors. According to Hoffman and Stuempfle, although there was a direct correlation between post-race sodium concentration and the frequency of sodium consumption by supplements, no difference in sodium intake was found between normo- and hyponatremic athletes [111]. In their study, they found that dietary sodium intake during training or competition for up to 30 h plays only a minor role in the development of EAH.

It has also been hypothesized by Hew-Butler et al. that sodium supplementation is not necessary for athletes, even during an Ironman that lasts more than 12 h [102]. The Institute of Medicine (USA) recommended a daily intake of sodium (1.5 g/65 mmol) and that appears to be sufficient for a healthy individual without the need for further supplementation during exercise [102]. In addition, the study by Barr et al. showed that salt intake during six hours of exercise does not correlate significantly with plasma sodium concentration [128]. Furthermore, the Institute of Medicine (USA) dietary recommendations state that for young adults, 1.5 g sodium (=65 mmol)/day, equivalent to 3.8 g NaCl) is sufficient to cover sodium sweat losses in non-acclimated individuals exposed to high temperatures or who are physically active [27]. Because the mean daily sodium intake in the United States is 3.1–4.7 g (135–204 mmol) for men and 2.3–3.1 g (100–135 mmol) for women, additional supplementation appears unnecessary in athletes following a Western dietary style [102]. Nevertheless, people who lose large amounts of sodium through sweat, such as competitive athletes or workers in extreme heat, should not consume the recommended intake of 1.5 g [17,76].

A number of factors contribute to the development of EAH, such as the sweat rate, and composition of the sweat, which makes it difficult to make definitive guidelines or recommendations regarding the type and/or quantity of fluid that should be consumed to prevent EAH [123]. A greater effort should be made to educate runners about valid techniques for monitoring their hydration status and developing appropriate individualized hydration strategies [129]. It is estimated that ~20% of marathoners require special sodium recommendations due to their high sweat salt losses [98]. Williams et al. conducted a survey evaluating participants from the London Marathon [130]. Even though 68.0% of participants had heard of hyponatremia or low sodium levels, only 35.5% knew its causes and effects [130].

These results highlight the need for educational interventions to increase the knowledge about hyponatremia and hydration strategies [131,132]. In addition, other climatic factors such as heat and humidity also play an important role for EAH that must be taken into account [106,133,134]. Furthermore, the interaction with carbohydrate intake must also be considered in this context. For example, studies have shown that carbohydrate absorption is enhanced by 30–50 mmol/L sodium, as glucose uptake in the intestine proceeds in co-transport with sodium. In addition, Barr et al. showed that 25 mmol/L sodium reduces plasma losses during exercise [106]. Similar recommendations can also be found from the sports medicine side in the context of Ironman events. In the publication of Pöttgen, the recommendation of 1–2 g of table salt (corresponding to 400–800 mg of sodium) per hour was found to prevent EAH and maintain plasma volume [135]. Thus, it can be stated that sodium intakes up to 800 mg/h in competition or 3 g/day as preparation, as in Koenders et al., bring no additional benefits in the worst case and enhance performance in the best case [27,100]. In addition to the amount of sodium, consideration should also be given to sensible amounts of fluids. For example, in their review, Holland et al. show that 0.14–0.27 mL of fluid per kilogram per hour (~588–1134 mL) improves endurance performance during exercise lasting > 2 h [136].

The drinking rates of 400–800 mL/h and 500–1000 mL/h recommended by Wharam et al. [85] and Speedy et al. [137] may also help to prevent EAH. It is recommended that fluid intake be customized in order to prevent excessive dehydration (>2% body weight) and body-weight gain [138]. In critical situations, hyponatremic runners who consume a hypertonic oral solution can quickly recover [73]. There is, however, a risk of fatal cerebral edema resulting from administering fluids to severely hyponatremic patients [139]. The patient’s sodium status should therefore be determined by laboratory tests prior to any treatment [140]. Further, SIAVP and hyponatremic encephalopathy might require alternative strategies, including 3% sodium chloride for the emergent treatment [141,142]. The onset of exercise-induced hyponatremia may be delayed.

Effective treatment to prevent hyponatremic states comprises oral hypertonic solutions that appear optimal, especially in combination with fluid restrictions. In addition, other treatment approaches, such as the administration of AVP antagonists or mannitol, have been tested. Despite this, the use of isotonic saline may not increase serum sodium levels, as previously shown in a study on marathon runners [143]. Hypertonic fluids may be used early in symptomatic patients [144]. If a hypernatremic collapsed runner is intolerant of oral fluid intake, intravenous fluid resuscitation may benefit them best [145]. However, collapse can also be caused by other factors requiring further diagnostics [146,147]. Furthermore, it may be possible to reduce the number of people who require medical treatment by adding a half-marathon and earlier start times [148]. Planners of the marathon should provide medical resources at the halfway point and at the last first-aid station [149,150]. Prior to the majority of runners reaching the middle- and later-distance checkpoints, resources and medical staff should be moved from the early tents to the later first-aid stations [149]. Further, pre-marathon evaluations and education might be beneficial to reduce medical complications [151].

## 5. Conclusions

Current data suggest that women with a low BMI and little competitive experience in endurance sports, such as marathon running, are at significant risk for EAH. The probability of developing EAH in a marathon is currently at ~8.5% [16,29,71,74,77], whereas renal dysfunction may occur in 8.7% of participants [16]. However, renal function abnormalities seem to be reversible and absent 2–6 days after a marathon run [152]. Various disciplines and race distances seem to be associated with different prevalence of EAH [82,123]. Due to the wide variation in sweat rates, renal water excretion, and the environment during race day, it is impossible to establish blanket guidelines regarding how much fluid should be consumed to prevent hyponatremia. But to avoid dehydration, especially during long efforts at high temperatures, and to optimize performance, drinking rates of 500–1000 mL/h should be aimed for but adjusted accordingly to individual losses. Since certain amounts of salt have not led to any negative consequences, the positive effects probably predominate here, even if these have not always been proven in the studies. In addition, with its association with improved carbohydrate intake, salt supplementation of 1–2 g/h is reasonable to exploit the positive potential [135,153]. A daily nutritional intake of up to 3 g sodium per day prior to a race has been found to positively affect athletic performance in many studies included in the present review [100,103,104,105]. Coaches of endurance athletes need to educate their athletes about the early symptoms of EAH to intervene at the earliest possible stage, although these symptoms are reported to be unspecific [70,154]. In addition, individual hydration strategies need to be developed during the daily training routine, ideally in combination with the determination of sweat rate and salt losses via sweat. According to Scheer and Hoffman, there is compelling evidence that drinking until one feels thirsty during endurance exercise is sufficient to maintain adequate hydration and prevent EAH [7]. However, athletes must be educated about the risks of over drinking in order to recognize even early symptoms of EAH [7]. They must strictly adhere to their individually composed hydration plan and should not drink beyond thirst sensation. Ultimately, this also has implications for organizers of multi-hour sporting events and their medical support staff. Organizers should consult with medical staff to determine where and at which locations beverage stations are appropriate and what beverages are ideally served. It is also important that the organizers inform the athletes at which refreshment stations they can obtain appropriate supplies so that they can adjust their individual supplies accordingly if necessary [155]. In addition, medical staff must be prepared for the fact that athletes may develop serious EAH depending on the duration of the event. This narrative review ultimately showed that hyponatremia is caused by a complex interaction of different factors. In any case, an important factor is the awareness of runners and coaches about this problem and the development of prevention strategies for training and competition.

## Figures and Tables

**Figure 1 jcm-11-06775-f001:**
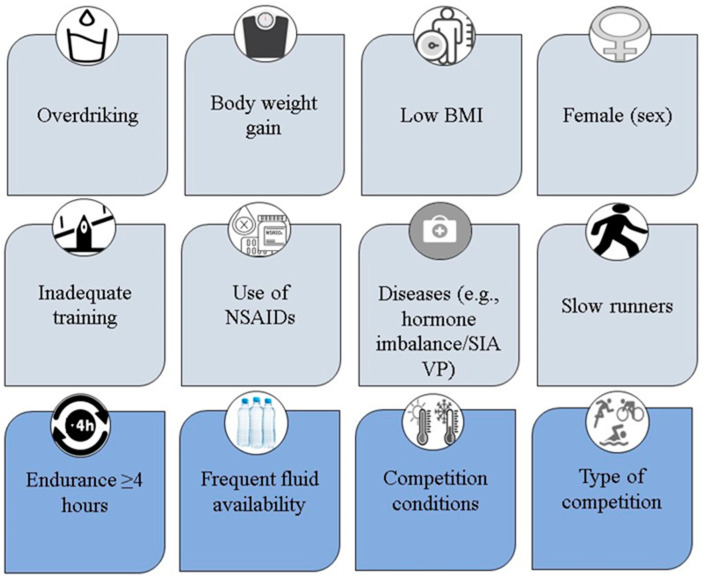
Illustration of person-related (potential) risk factors and event-related risk factors for exercise-associated hyponatremia. BMI, body mass index; NSAIDs, Non-steroidal anti-inflammatory drugs; SIAVP, Syndrome of inappropriate antidiuretic hormone secretion.

**Table 1 jcm-11-06775-t001:** Definitions of different forms of hyponatremia [46,47].

EAH Form	Definition
Pseudohyponatremia	Plasma’s lipid, protein, and glucose content are increased, whereas the sodium content of plasma water is within the normal range (135–145 mmol/L)
Euvolemic hyponatremia	Total body volume increased, whereas total sodium is normal (135–145 mmol/L)
Hypovolemic hyponatremia	Total body volume decreased and serum sodium < 135 mmol/L
Hypervolemic hyponatremia	Total body volume increased and serum sodium < 135 mmol/L
Biochemical hyponatremia	Serum sodium ranges from 129 to 134.9 mmol/L
Clinically significant hyponatremia	Serum sodium < 129 mmol/L

**Table 2 jcm-11-06775-t002:** Prevalence of exercise-induced hyponatremia (EAH) in marathon runners.

Study (Year)	Prevalence of EAH
Hew et al., 2003 [69]	<1%
Chorley et al., 2007 [70]	22%
Almond et al., 2005 [71]	13%
Hsieh et al., 2002 [72]	5.6%
Siegel et al., 2009 [73]	4.8%
Mettler et al., 2008 [74]	3%

**Table 3 jcm-11-06775-t003:** Hydration characteristics of participants in the studies from Noakes et al. [47] and Glace et al. [75].

Author (Year)	Overhydrated		Euhydrated		Dehydrated		Correlation between Body Weight and Serum Sodium
	Biochemical hyponatremia	Clinically noticeable hyponatremia	Biochemical hyponatremia	Clinically noticeable hyponatremia	Biochemical hyponatremia	Clinically noticeable hyponatremia	
Noakes et al., 2005 [47]	44	25	41	6	38	0	inverse
Glace et al., 2022 [75]	1	0	0	0	0	0	inverse

## Data Availability

Not applicable.

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
