# Peer review of "Exercise-Associated Hyponatremia in Marathon Runners"

_jcm, 2022, doi:10.3390/jcm11226775_

Round 1
Reviewer 1 Report
Klingert et al. prepared a review titled “Exercise-Associated Hyponatremia in Marathon Runners”. The authors provide a nice overview about many aspects of hyponatremia in long distance running. The literature search seems appropriate. The manuscript is well written and clear. Overall, the review summarizes the existing literature well.
Here are some comments:
1. Abstract/Line 18: What is ‘its’ referring here?
2. Line 49: This is just the case for the excessive intake of low-sodium or hypotonic fluids.
3. Please discuss also potential dangers of overdosing sodium-rich supplements?
4. Line 114: ‘onset’ of what?
5. Please cite or indicate whether there are similar reviews available. This might be helpful for interested readers.
6. With the limited information, Table 3 is difficult to understand. What do the numbers stand for?
7. Line 251: “It has 251 been found that serum sodium concentration and body weight change were significantly 252 correlated.” Inversely correlated?
8. Many of the works you are talking about discuss the problem of excessive fluid consumption. Did any of these studies distinguish between the type of fluids (hypotonic, isotonic, hypertonic) and their impact on sodium levels? Could there be a weight gain by drinking isotonic (or hypertonic) fluids, without increasing the risk for hyponatremia? This is especially important for the statement starting from Line 606 to 611.
9. This might be speculative, but isn’t there an increasing awareness of the problem of hyponatremia (or water intoxication), which could make it less common?
10. Table 4 is a figure.
11. You discuss the potential benefits of sodium supplementation. Could you briefly state potential risks of overdoing the sodium intake?
Author Response
Reviewer 1
Comments and Suggestions for Authors
Klingert et al. prepared a review titled “Exercise-Associated Hyponatremia in Marathon Runners”. The authors provide a nice overview about many aspects of hyponatremia in long distance running. The literature search seems appropriate. The manuscript is well written and clear. Overall, the review summarizes the existing literature well.
Here are some comments:
- Abstract/Line 18: What is ‘its’ referring here?
Answer: ‘Its’ in this sentence referred to exercise-associated hyponatremia. The sentence has been clarified.
- Line 49: This is just the case for the excessive intake of low-sodium or hypotonic fluids.
Answer: Thank you for this statement. The sentence has been revised and changed according to the details in the cited publication.
- Please discuss also potential dangers of overdosing sodium-rich supplements?
Answer: We thank the reviewer for this advice. We could not find a review or specific article discussing the negative effects and dangers of overdosing sodium-rich supplements in marathon runners. However, a sentence has been added pointing out risks of sodium overdose in general (lines 71-72).
- Line 114: ‘onset’ of what?
Answer: We thank the reviewer for this question. The sentence has been clarified.
- Please cite or indicate whether there are similar reviews available. This might be helpful for interested readers.
Answer: We thank the reviewer for this suggestion and referred to a review focusing on exercise-associated hyponatremia in general (line 75).
- With the limited information, Table 3 is difficult to understand. What do the numbers stand for?
Answer: The numbers in Table 3 indicate the number of runners. The heading has been adapted accordingly.
- Line 251: “It has 251 been found that serum sodium concentration and body weight change were significantly 252 correlated.” Inversely correlated?
Answer: We thank the reviewer for this question. The sentence is indeed correct and the cited publication states: “The finding of a statistically significant correlation between postrace serum sodium concentration and change in body weight in runners confirms the findings of other authors.” (Reid et al., 2004).
- Many of the works you are talking about discuss the problem of excessive fluid consumption. Did any of these studies distinguish between the type of fluids (hypotonic, isotonic, hypertonic) and their impact on sodium levels? Could there be a weight gain by drinking isotonic (or hypertonic) fluids, without increasing the risk for hyponatremia? This is especially important for the statement starting from Line 606 to 611.
Answer: We thank the reviewer for this question. Most of the studies did not distinguish between the type of the consumed fluids. Furthermore, there is evidence that the incidence of exercise-associated hyponatremia is associated with inappropriate physiological mechanisms rather than the type of fluid consumed.
- This might be speculative, but isn’t there an increasing awareness of the problem of hyponatremia (or water intoxication), which could make it less common?
Answer: Even though there is an increasing awareness of the problem of hyponatremia, the incidence in marathon runners is still high. In addition, recommendations for appropriate fluid intake have been adapted to recent research findings and need to be communicated to raise the awareness of the runners.
- Table 4 is a figure.
Answer: We thank the reviewer for the hint and have changed the wording to Figure.
- You discuss the potential benefits of sodium supplementation. Could you briefly state potential risks of overdoing the sodium intake?
Answer: See answer to comment #3: We thank the reviewer for this advice. We could not find a review or specific article discussing the negative effects and dangers of overdosing sodium-rich supplements in marathon runners. However, a sentence has been added pointing out risks of sodium overdose in general (lines 71-72).
Reviewer 2 Report
I would like to thank the authors for the effort to write such a comprehensive review article on a little-studied topic on Excercise-associated Hyponatremia in Marathon Runners.
The article provides relevant and interesting information on the most frequent hydroelectrolytic alteration in marathon runners, and that, despite recent research, the real physio-pathogenic mechanisms that generate it are still not known.
The manuscript correctly describes the introduction, materials and methods, results and discussion of this very interesting topic, as well as a very detailed description of all related work on the subject.
However, I would like to express certain aspects that I will comment below for the consideration of the authors:
-In lines 48 and 52, the sentences are repeated, even though you provide different bibliographic citations, please unify them.
-On line 56, please describe the acronym TCS.
-In lines 114 to 122, the authors describe the causes of hyponatremia appearing at 48hrs, I do not understand adding this comment when it is often not related to the EAH issue. I recommend deleting it or try to do so by associating it to the topic.
Author Response
Reviewer 2
Comments and Suggestions for Authors
I would like to thank the authors for the effort to write such a comprehensive review article on a little-studied topic on Excercise-associated Hyponatremia in Marathon Runners.
The article provides relevant and interesting information on the most frequent hydroelectrolytic alteration in marathon runners, and that, despite recent research, the real physio-pathogenic mechanisms that generate it are still not known.
The manuscript correctly describes the introduction, materials and methods, results and discussion of this very interesting topic, as well as a very detailed description of all related work on the subject.
However, I would like to express certain aspects that I will comment below for the consideration of the authors:
- In lines 48 and 52, the sentences are repeated, even though you provide different bibliographic citations, please unify them.
Answer: We thank the reviewer for this comment and have unified both sentences.
- On line 56, please describe the acronym TCS.
Answer: We thank the reviewer for this comment and have indicated the abbreviation.
- In lines 114 to 122, the authors describe the causes of hyponatremia appearing at 48hrs, I do not understand adding this comment when it is often not related to the EAH issue. I recommend deleting it or try to do so by associating it to the topic.
Answer: We thank the reviewer for this comment. The paragraph has been added to provide some background details on hyponatremia and medical impacts. A sentence has been added and the paragraph has been adjusted to associate it to the topic.